# Evaluation of Food Labelling the Products with Information Regarding the Level of Sugar: A Preliminary Study

**DOI:** 10.3390/nu14132697

**Published:** 2022-06-29

**Authors:** Marta Sajdakowska, Jerzy Gębski, Aleksandra Wardaszka, Anita Wieczorek

**Affiliations:** Department of Food Market and Consumer Research, Institute of Human Nutrition Sciences, Warsaw University of Life Sciences (WULS-SGGW), 159C Nowoursynowska Street, 02-776 Warsaw, Poland; jerzy_gebski@sggw.edu.pl (J.G.); aleksandrawardaszka@gmail.com (A.W.); aaanita.wieczorek1997@gmail.com (A.W.)

**Keywords:** consumers, information on food label

## Abstract

The aim of this article is to explore the reasons for seeking selected information on a food label, with particular emphasis on certain information about sugar. In order to meet the aim, in 2020, a survey was conducted among consumers aged 18–45. Predictive models (Logistic Regression) were developed for noticing “light” products and reading food labels. The results of our study indicate that consumers are mainly discouraged from reading label information by a large amount of information, the lack of time, and a general reluctance to be interested in label information. When it comes to the factors that lead people to read label information, the naturalness of the product, its organic origin, and physical activity are important. Moreover, respondents who declared that they noticed products on the market defined as “light” were those who were interested in the naturalness of the product, but also consumers declaring that they have nutritional knowledge. The results of the study indicate the need to intensify information campaigns in order to emphasize how important it is for consumers to read the information on food labels. The amount and complexity of information currently appearing on the food label may unfortunately discourage consumers from reading it, so it is important to continuously improve this form of communication with the consumer.

## 1. Introduction

Searching for food products on the market, consumers use various sources of information, including the information on the food label [1,2,3]. The label of a food product plays a significant role, as it provides all the mandatory information regarding the nutritional composition, safety, and quality of food. Basically, labels provide information on ingredients of the food products, nutritional properties, preparation, and storage. Decisions mainly depend on the perception of the product, and food labelling is one of the most important factors affecting the purchasing decisions of consumers [4]. Thus, nutrition labelling may be an effective approach to empowering consumers in choosing healthier products [5,6]. It is also important to remember that consumers may not use nutritional labels because they require time and effort to process the information [7].

Research into the literature indicates that the prevalence of diet-related diseases continues to rise in the U.S. and globally [8]. Therefore, the right way to provide information to the consumer becomes especially important because poor dietary patterns, high energy intake, and malnutrition are some of the major causes of non-communicable diseases (NCDs) such as obesity, diabetes, cardiovascular disease, and some types of cancer [9].

Furthermore, due to the growing interest in health and wellbeing, in general, the food market introduced some types of food enrichment by adding beneficial ingredients to foods (calcium, fibre) [10] or proposed the reduction or elimination of specific components (e.g., sugar) with the purpose of reducing the calorie intake [11,12]. These modifications are communicated through claims on the package. In 2006, the first European regulation specifically addressing nutrition and health claims was introduced to avoid misunderstanding and to protect consumers against false information [13].

Studies in the literature indicate that, in general, consumers who are health-conscious with respect to their lifestyle and diets derive high utility values from the nutritional information of the product [14,15]. Middle-aged people [16,17] and those who are better educated [18,19] are more likely to seek information on the label. Education can increase nutritional knowledge and strengthen the health motivation; the former is known to boost understanding and the latter to increase label usage [15]. Additionally, as noted earlier, the information on the packaging/label is very important for people who have a special interest in health issues [18,20], as well as for people for whom, for example, the naturalness of food is important or its organic origin [21,22]. On the other hand, research also indicates that consumer groups with low health orientation (specifically smokers, those who do not exercise regularly, and those with an unhealthy body weight) show little interest in nutritional labels [16]. Research indicates that having nutritional knowledge is likely to help by directing attention to salient information, promoting comprehension and allowing more accurate information to be stored in memory and used in decision-making situations [23]. Additionally, shoppers exposed to front-of-package (FOP) labels had an increased intention to purchase healthier foods [24]. For some ingredients that are important not only for their health effects, but also for a person’s well-being, including their appearance (e.g., sugar increases the likelihood of being overweight or obese), consumers may pay particular attention to their presence in the diet [25,26]. Additionally, at the same time, they can thus look for information about them on the packaging/product label [26,27].

However, apart from consumer behaviours referring to a healthy lifestyle, there are also behaviours that should definitely be modified [28,29,30]. This is even more necessary due to the fact that there is a continuous increase in the consumption of products containing relatively large amounts of added sugar [31,32,33]. In Poland, it was reported that in 2008–2017, the annual consumption of unprocessed sugar decreased by 5.7 kg, but the total annual consumption of sugar in forms other than unprocessed sugar increased by 11.8 kg [34]). Other studies indicate that the consumption of unprocessed sugar in 2018 amounted to 0.94 kg/person/month and was lower than that recorded in 2010 (1.30 kg/person/month) by almost 30%, however, but it is worrying that the consumption of confectionery products increased by about 20% over the period (from 0.39 kg/person/month in 2010 to 0.48 kg/person/month) [35]. Additionally, in the context of health effects, consumer interest in, e.g., carbonated, energy, and isotonic drinks [34], which also introduce additional sugar into the diet, is reported [36].

On the one hand, the growing interest in healthy lifestyles among certain groups of consumers, and on the other, the choice of food products preferred for their convenience or taste, makes it necessary to seek ways to encourage consumers to make more informed decisions [37,38]. As noted earlier, food labelling/packaging can be a helpful tool in these market choices. Therefore, the aim of the article is to explore the reasons for looking for selected information on a food label, with a particular emphasis on certain information about sugar.

## 2. Materials and Methods

### 2.1. Data Collection Process

Research was performed via an online survey. To assess the usefulness of food labelling information for consumers, a specific online questionnaire was designed on Google forms. Prior to the survey, all of the respondents were told that the data and opinions gathered through the survey were confidential and would be used for research purposes only. The survey distribution was mainly performed by email invitation and social media for a period of 3 months (August–October) in 2020 only among people who expressed their willingness to participate in the study. For this purpose, appropriate information was provided to survey participants, allowing them to decide their participation in this research study. Only people who declared that they were 18 years of age or older participated in the study. In the end, 270 people aged 18–45 joined the study, out of which 265 respondents meeting the study criteria were classified for analysis, i.e., declaring participation or co-participation in food shopping in the household.

The study assessed the interest in the information on the label of the food product, with particular emphasis on the products referred to as “light”. For the purposes of the research, the research questionnaire presents the definition of “light” products, describing them as “products that are characterized by a reduced content of one or more nutrients (e.g., fat, sugar) or a lower calorific value”. The survey also took into account perceptions of information about the presence or addition of sugar in foods.

### 2.2. Description of Questionnaire

In order to obtain the opinion on the importance of information on food label/packaging, an original survey questionnaire was designed, which contained over a dozen questions, including questions regarding demographic variables. Generally, the questions concerned (a) the interest in information on food packaging and related to selected information on the presence or reduced content of sugar in food products as well as (b) the questions concerned the determination of projective image of buyers. The questionnaire also included statements derived from the consumer innovativeness scale [39], which has been used in other studies conducted on the food market [40,41]. The questionnaire used in the study consisted of the following questions:

(1) Do you read food labels? (filtering question; Yes/No)

If you read the information, what information on the packaging do you pay attention to? (you can choose 3 answers from the given ones)

–Calorific value of the product.–Sugar content.–Fat content.–Protein content.–Addition of enhancers, preservatives.–Vitamin and mineral content.

If the respondent declared the answer that they do not read, they moved on to the question: 

(2) Why do you not read food labels? (After answering this question, the respondent is automatically redirected to the end of the survey) (There is a choice of 3 answers from the given ones).

–I do not have time to read the labels.–I do not feel the need to know what is in the product.–I do not feel like reading them.–The font on the label is too small, making it uncomfortable to read.–I am not reading them because there is too much information on the label.

Subsequently, the subjects learned the definition of “light” products: products that are “light” are characterized by a reduced content of one or more nutrients (e.g., fat, sugar) or by a lower calorific value.

After reading the definition, respondents answered the following questions.

(3) Have you noticed any “light” products on the market? (Yes/No).

(4) Which product would you choose when shopping if you had a choice of products with the following information on the label? (Question with 5-point scale, where 1—I agree least and 5—I agree most).

–“Zero sugar”.–“No added sugars”.–“Does not contain sugar”.–“Reduced sugar content”.–“Contains only naturally occurring sugars” (e.g., from fruit).

(5) Please indicate how much you agree with the statement that people who are aware of the consequences of too much sugar consumption include (question with 5-point scale where 1—I agree least and 5—I agree most):–Young adults.–People knowledgeable about nutrition.–Physically active people.–Busy people.–People with health problems.–Professionally active people.–Elderly people.–People looking for nutritional novelties.–Wealthy people.–People who like cooking.–Single people.–Convenience-oriented people related to the preparation of a meal.–People looking for bargains.–Families with children.–People who care in a special way about their health.

(6) How would you rate yourself in terms of lifestyle? (Question with 5-point scale, where 1—I agree least and 5—I agree most). Please indicate how much you agree with the end of the sentence: I believe that I am a person...:–Caring about my own health.–Committed to my professional work.–Attentive to the naturalness of food.–Family-oriented.–Valuing tradition.–Environmentally conscious.–Physically active.

### 2.3. Statistical Analysis

Preliminary analysis of the results included socio-demographic characterization of the sample through frequency analysis for qualitative questions and assessment of the mean for quantitative questions (quantitative variable, qualitative variable).

A comparison was also made of the average values of the obtained responses, broken down into categories related to noticing light products on the market: “Have you noticed any “light” products on the market?”. Due to the binary nature of the categorical variable, Student’s *t*-test for independent groups was used.

Predictive models were subsequently developed for noticing light products and reading food labels. Logistic Regression models were used in both models due to the dichotomous nature of the dependent variable.

For the prediction of noticing light products, the dependent variable was the categorical variable used in Student’s *t*-test. The prediction of the probability of obtaining the answer “Yes” was made. The explanatory (independent) variables in this model were questions about awareness of the consequences of consuming sugar, variables describing lifestyle, and demographic variables: gender, age, place of residence, education, occupational situation, subjective opinion about income, number of people in the household, and number of children. The following explanatory variables describing lifestyle and “metric” variables were used in the model to predict the likelihood of reading a product label: gender, age, place of residence, education, occupational situation, subjective opinion of income, number of people in the household, and number of children.

Good fit of both models was confirmed (Model Fit Statistics) by the Hosmer and Lemeshow Goodness-of-Fit Test (respectively: model for label *p* = 0.8793 and model for information on sugar/light *p* = 0.6143). The statistical analysis was performed using the SAS 9.4 statistical package (SAS Institute, Cary, NC, USA).

## 3. Results

### 3.1. Description of the Sample

Questions concerning socio-demographic characteristics such as gender, age, education, subjective assessment of the financial situation, place of residence, declaration of purchase of health-promoting products were also included. Details of the studied sample are presented in Table 1. Women predominated in the study (86.8%). There were slightly more people under the age of 30 (55.1%) compared to people aged 31–45 and people with a higher education (59.3%). More than 40% of the respondents lived in rural areas (44.5%). About 70% of the respondents assessed their financial situation at an average level (respectively, we can afford some, but not all expenses—38.9%; we can afford everything—32.8). About 1/3 of the respondents (34.3%) declared that they buy health-promoting products quickly, although after some consideration, and almost 1/3 (27.2%) that they do so only after some friends have already tried it (Table 1).

### 3.2. Opinions on Reading Information on the Food Label, Including Information on Sugar

Among the survey participants (265), 78 people said they do not read labels and 178 said they read labels. The respondents who claimed not to read labels were asked why they were not interested in the information on the food packaging. The main reasons for not reading the information on the label were: lack of time (37.2%), unwillingness to read (“I do not feel like reading”) (30.8%), too much information (30.8%), no need to know what is in the product (20.5%), but also inappropriate font size (14.1%) (Table 2).

People who declared that they read labels were asked to indicate what information they pay attention to, with particular emphasis on information related to the composition of the food product. In terms of information sought on the product label, the highest percentage of indications was sugar content (75.4%), calorific value (65.2%), addition of enhancers/preservatives (56.7%), followed by fat content (31.5%). The rest of the ingredients received percentages of about 12% and lower (protein content; vitamin and mineral content) (Table 3).

Data analysis indicated that socio-demographic variables were not statistically significant in a model that was designed to assess the effect on reading/seeking information on a food label.

However, it was noted that self-reported lifestyle assessment and the level of interest in new products on the market with health-promoting features played a role (Table 4). The dependent variable (explained, regressor) is the dichotomous variable of reading food labels. The probability of an affirmative “Yes” response is modelled. The independent variables (explanatory, regressors) are variables characterizing the lifestyle of the respondents.

The first statistically significant variable in the model is attention to food naturalness. An increase in attachment to food naturalness resulted in an increase in the likelihood of being a food label reader. Each 1-point increase in the rating of this trait resulted in an increase in the likelihood of reading food labels by more than 150% (OR: 2.529; 95% CI: 1.59–4.01).

The second significantly statistically significant variable in the model is the respondents’ high environmental awareness, and an increase in this self-assessment resulted in a 97% increase in the odds of reading food labels for each point of this assessment (OR: 1.971; 95% CI: 1.27–3.07).

The respondents’ physical activity had a positive effect on the likelihood of reading food labels. Each 1-point increase in physical activity in the respondent’s opinion increased the odds of being a food label reader by 12% (OR: 1.120; 95% CI: 1.07–1.43).

The level of “family orientation” had a negative effect on the probability of reading food label content. Each increase in the level of this variable resulted in a decrease in the probability of reading the label by 32.5% (OR: 0.675; 95% CI: 0.43–0.86).

Another variable with a statistically significant effect on the model is attachment to tradition. An increase in awareness of this trait also had a negative effect on the likelihood of reading food labels. A 1-point increase in this opinion resulted in a 38.1% decrease in the probability of reading labels (OR: 0.619; 95% CI: 0.41–0.93).

For interest in new food products (qualitative variable), an increase in interest in new products was closely related to an increase in the likelihood of label reading. Individuals who immediately purchase a food product when it appears on the market were 5.66 times more likely to read the label (OR: 5.658; 95% CI: 2.73–12.85) than those who were reluctant to purchase new health-promoting foods (reference level). People who buy food products quickly after market introduction, but after some consideration, had a 3.92 times greater chance of reading the product label than the reference level (OR: 3.918; 95% CI: 1.51–10.20). For people who buy a new product only after some friends have tried it, the chance of reading the label is 80% greater than the reference level (OR: 1.795; 95% CI: 1.07–4.51). For people who buy a new food product only after most of their friends have already bought it and reviewed it positively, this chance is 25% higher than for people belonging to the reference level (OR: 1.250; 95% CI: 0.47–3.33); however, for this level, there is no statistical significance for the model (Table 4).

The dependent variable (explained, regressor) is the dichotomous variable of noticing light products on the market. The probability of an affirmative answer “Yes” is modelled. The independent variables (explanatory, regressors) are two variables characterizing the lifestyles of the respondents (awareness of the consequences of high sugar intake among those with nutritional knowledge and paying attention to the naturalness of the food consumed). Consumers who were aware of the consequences of high sugar intake, claiming to be nutritionally knowledgeable (declaring to be aware of the consequences of high sugar intake), were 256% more likely to notice light products on the market for each point increase in self-assessment of this trait (OR: 3.566; 95% CI: 1.29–9.83). People who paid attention to the naturalness of food were five times more likely to notice light products for every point increase in self-assessment of this trait (Table 5).

### 3.3. The Importance of Information on Sugar in Product Choice and Perception of the Consumer Purchasing Foods with Reduced Sugar Content

In terms of product choice depending on the information on the packaging/label regarding the sugar content of the product, respondents who declared that they noticed light products on the market were statistically significantly more likely to agree that they would choose products with the information “Contains only naturally occurring sugars”, “No added sugars”, “Reduced sugar content”, “Sugar free”, and “Zero sugar” compared to respondents who said they did not notice light products on the market (Table 6).

In terms of agreement with the statements characterizing people who are aware of the consequences of too high a sugar intake, people declaring that they had noticed light-type products compared to people who declared that they had not seen such products on the market were statistically significantly more likely to say that this group includes “people knowledgeable about nutrition” (4.5), people who are physically active (4.3), and people who take special care of their health (4.2).

It was also pointed out that such people include people seeking food novelties (3.9), as well as people who enjoy cooking (3.6), people with health problems (3.6), as well as people who are perceived as wealthy (3.4) (Table 7).

## 4. Discussion

### 4.1. Factors Determining Disinterest in Label Information and Encouraging Label Reading

The results of our study indicated that consumers are mainly discouraged from reading label/packaging information by the large amount of information, lack of time, and general reluctance to be interested in label information. This is confirmed by the findings of other authors, because the results show that consumers usually do not read food labels due to lack of time [1] and excessive information [1,42]. Furthermore, a cost is involved in reading and processing any food label. For each consumer, the cost of not reading a food label is likely to be quite low, but across a large number of purchasers, it might turn out to be significant [43]. Moreover, it is important to pay attention not only to the type of information placed on the labels but also its quantity, and last but not least, credibility [44]. In addition, efficiency in labelling relies on widespread uptake in the marketplace, and quality communication and consumer education [45]. The literature research further indicates that there are different groups of factors that are important in information seeking. In general, the framework mainly focuses on personal factors; that is, how consumers perceive and process information, the antecedents of that information, and how they create a tendency to search for additional label information. These personal factors have been organized into three major groups: general personal factors (e.g., health consciousness and socio-demographics), label-related personal factors (e.g., label-related self-efficacy, trust in labels, and the perceived usefulness of labels) and product-category-related personal factors (trust in food products, enduring involvement, experience, and perceived quality differences) [46]. Other research indicates that some consumers may perceive difficulty with the presence or absence of certain ingredients in a product and the general interpretation of label information. Evidence from the study of Hartmann et al. (2018) implies that free-from labels might confuse certain consumers, which might encourage the use of a simple heuristic to evaluate different food options: labelled is better than unlabelled. With the growing number of food labels, consumers need to be informed about their correct meanings and interpretations in order to prevent misconceptions and unwanted compensatory effects on food intake [47].

Generally, when shaping consumers’ food shopping behaviour using certain food labels, it is important to know how consumers interpret those labels. A false interpretation of labels might lead to unintended changes in consumer behaviour [47]. Additionally, food labelling was observed to be more useful for specific consumer groups, e.g., consumers with health conditions or consumers concerned with a healthy lifestyle [1]. Moreover, Plasek et al. (2020) yielded six separate categories that influence consumers in their perception of the healthiness of food items: (1) the communicated information—such as FOP (front-of-package) labels and health claims, (2) the product category, (3) the shape and colour of the product packaging, (4) the ingredients of the product, (5) the organic origin of the product, (6) and the taste and other sensory features of the product [48].

Our research indicates that the factors that lead people to seek/read label information are the naturalness of the product, its organic origin, and physical activity, which is supported by other studies [21,22,42]. Literature studies indicate that environmental aspects are important to consumers [49,50]. In addition, with regard to food labelling, the information on environmental intervention and its joint use with green identity labels can enhance consumer valuation for already existing pro-environmental foods [51]. The literature also indicates that consumers with active sport lifestyles showed higher awareness of nutritional information and a higher level of nutritional knowledge in relation to their attitudes toward other information [42]. Furthermore, health evaluation, information-seeking, nutrition knowledge, and preference for naturalness predicted intention to pay a premium price for some products [47]. Moreover, the research results indicate that the information on the package has the potential to encourage healthier purchasing and potentially improve the diet quality of consumers [52].

Our study indicated that an increase in interest in products perceived by consumers as new to the market was closely related to an increase in the likelihood of label reading. The literature indicates that selected factors such as innovation-related perception, customer perceived value, as well as customer perceived risk, are all important variables related to the acceptance of an innovative food product [53]. Moreover, research results related to the ingredients of the product confirm that reducing the sodium, sugar, and fat content increases consumer acceptance of the improved product in terms of health [48].

### 4.2. Information about Sugar on the Label in the Opinion of the Respondents

Reduction in sugar intake of some product is advised around the world as part of healthier dietary patterns to help reduce energy intakes, obesity risk, and obesity-related disorders [54]. Other literature studies also indicate that for some food groups, sugar content reduction could have the greatest impact on public health [55]. As far as Polish consumers are concerned, it seems particularly important that the information on the label (including information about sugar content in the product) is presented in an appropriate way so that the label can be properly interpreted and understood by the consumer, especially in the situation of a worrying increase in the number of overweight and obese people in recent years [56].

The results of our study indicate that those who declared that they noticed products on the market defined as light (which may be perceived by consumers not necessarily as new, but in which there is a change consisting in lowering the sugar level in the product formulation) included those who were interested in the naturalness of the product, but also consumers declaring that they have nutritional knowledge. The aspect of the naturalness of the product and not interfering externally/artificially with its recipe (e.g., not adding this ingredient, e.g., sugar during, e.g., the production process) are confirmed by consumers’ opinions indicating that they are more likely to choose products whose labels include such wording (statements with the highest average “Naturally occurring sugars” and “No added sugars”). The literature confirms that some consumers are looking for products without added sugar or with naturally occurring sugar in the product [25,26,27].

In our research, the importance of health preferences and physical activity is confirmed by the perceived profile of people who were considered to be aware of the consequences of high sugar intake. According to the consumers surveyed, these people included mainly those with nutritional knowledge, those who are physically active and take special care of their health, but also those who look for nutritional novelties.

The literature indicates that, in general, healthy dietary patterns are more likely to coexist with self-reported moderate and high physical activity during work/school and leisure time [57]. Moreover, decisions made by participants concerning the healthfulness of food products were significantly influenced by sugar content [58]. In addition, consumers concerned about their diet (i.e., proper diet) may seek products with lower sugar content [59]. Research also indicates that consumers have their own developed ways of lowering/reducing the amount of sugar in their diets, e.g., research using focus group interviews indicates that participants revealed some strategies they found to work well for them (or for people they know) when trying to reduce their amount of sugar intake. These included: reducing the amount of consumed sugar gradually, while maintaining some flexibility (i.e., allowing themselves some occasions where they could eat some sweets); choosing and/or buying products with lower sugar content; trying to cook at home more frequently and adding less sugar to recipes; not adding sugar to coffee or tea; substituting sweet desserts with fruit; and establishing a weekly maximum intake of specific products [60]. However, it is still recommended to increase the level of nutrition knowledge among consumers in order to better understand the information on the label and consequently make better choices on the food market [15].

In addition, due to the excessive consumption of sugar by today’s consumers, there are activities in the market to reduce the amount of sugar in food products, but companies should strive to reduce the amount of sugar in food products to a greater extent [61]. With regard to sugar content, it is useful to identify specific groups of food products in which sugar reduction can have a significant impact on public health. Therefore, appropriate labelling can be/become an adequate tool to implement and evaluate actions aimed at reducing sugar consumption [55]. The literature indicates that, with respect to the sugar content of foods, in addition to the commonly consumer-recognized means of labelling foods, other methods of labelling are also proposed to allow the consumer to better estimate or understand how much sugar is in a food product, e.g., indicating the amount of sugar in teaspoons, which, as a result, may contribute to the likelihood of purchasing decisions that are more rational from a health perspective [62].

### 4.3. Practical Implications, Strengths, and Limitations

This article is not without limitations. Therefore, in future studies, we propose an increase in the study sample and/or examination of how consumers evaluate label information in relation to other nutrients, because sugar is not the only key ingredient that consumers must evaluate to make purchase choices.

Despite these limitations of the study, we hope that our findings can provide practical implications. There is an increasing segment of individuals who aim at a healthier way of living, and show that by making healthier food choices. For this segment, it is important that the healthiness of food items is clearly communicated, allowing them to make better choices.

Furthermore, it seems that in addition to the educational campaigns on healthy eating habits and the importance of physical activity, which have been conducted in Poland so far and have been aimed at children and young people, it is also worth remembering to undertake educational campaigns aimed at other age groups (i.e., young, middle aged, and older consumers). It seems important to emphasize the need to reduce sugar in the diet for health reasons, among others. In 2021, following the example of other European and world countries, a sugar-sweetened beverage tax was introduced in Poland to reduce the consumption of free sugars in beverages. However, it will take time to assess to what extent this legislative change will promote healthier consumer choices.

## 5. Conclusions

The results of the study indicate the need to intensify information campaigns in order to emphasize how important it is for consumers to read the information on food labels. It should also be emphasized that the amount and complexity of information currently appearing on the food label may unfortunately discourage consumers from reading it. Therefore, it is important to continuously improve this form of communication for the consumer, especially now that further attempts are being made in Europe to develop a more convenient system for communicating information contained on the food label. The primary goal should be to develop such an approach that allows the communication of the necessary information, which should be visible, but also correctly understood by consumers.

## Figures and Tables

**Table 1 nutrients-14-02697-t001:** Socio-demographic characteristics of the study sample (N; %).

		Frequency (N)	Percent (%)
Gender	Female	230	86.8
	Male	35	13.2
Age	18–30	146	55.1
	31–45	119	44.9
Education	Secondary and lower	108	40.7
	Higher	157	59.3
Place of residence	Countryside	118	44.5
	City up to 50,000 inhabitants	51	19.3
	City of 50–250 thousand inhabitants	35	13.2
	City with more than 500,000 inhabitants	61	23.0
Opinion on family income	It is not sufficient at all or it allows to satisfy only basic needs	31	11.7
	We can afford some, but not all, expenses	103	38.9
	We can afford everything	87	32.8
	We can afford everything, plus we can save	44	16.6
I buy health-promoting products	As soon as they become available	8	3.0
	Quickly, though after some consideration	91	34.3
	Only after some friends have already tried them	72	27.2
	When most of my friends have already bought and reviewed them positively	46	17.4
	I am reluctant to buy new foods with health-promoting benefits	48	18.1

**Table 2 nutrients-14-02697-t002:** Reasons for not reading label information (N; %).

Statements	N	%
I do not have time to read the label	29	37.2
I do not feel the need to know what is in the product	16	20.5
I do not feel like reading it	24	30.8
The font on the label is too small, making it uncomfortable to read	11	14.1
I am not reading because there is too much information on the label	24	30.8

**Table 3 nutrients-14-02697-t003:** Types of information sought on a product label (N; %).

Information	N	%
Calorific value of the product	122	65.2
Sugar content	141	75.4
Fat content	59	31.5
Protein content	23	12.3
Addition of enhancers/preservatives	106	56.7
Vitamin and mineral content	12	6.4

**Table 4 nutrients-14-02697-t004:** Declaration of reading food labels (YES/NO).

Variable	Label	Estimate	Point Estimate	95% Wald Confidence Limits		*p*-Value
Intercept		−0.966				0.2057
How would you rate yourself in terms of lifestyle?	I draw attention to the naturalness of food	0.928	2.529	1.59	4.01	<0.0001
	I am family-oriented	−0.393	0.675	0.43	0.86	0.0359
	I value tradition	−0.479	0.619	0.41	0.93	0.0194
	I am environmentally conscious	0.678	1.971	1.26	3.07	0.0027
	I am physically active	0.113	1.120	1.07	1.43	0.0372
How do you buy a new health-promoting food product?	As soon as they become available	1.733	5.658	2.73	12.85	0.0481
	Quickly, though after some consideration	1.366	3.918	1.50	10.20	0.0052
	Only after some friends have already tried them	0.585	1.795	1.07	4.51	0.0212
	When most of my friends have already bought and reviewed them positively	0.223	1.250	0.47	3.33	0.6556
	I am reluctant to buy new foods with health-promoting benefits (ref.)	0	1			.

Point estimate—OR (e^β^); (95% Cl)—95% Wald Confidence Limits; ref.—reference group.

**Table 5 nutrients-14-02697-t005:** Declaration of noticing “light” products on the market.

Variable	Label	Estimate	Point Estimate	95% Wald Confidence Limits		*p*-Value
Intercept		−4.975				0.0363
Aware of the consequences of high sugar intake	Knowledgeable about nutrition	1.2714	3.57	1.29	9.83	0.014
How would you rate yourself in terms of lifestyle?	Paying attention to the naturalness of food	1.6223	5.06	1.15	22.35	0.0322

Point estimate—OR (e^β^); (95% Cl)—95% Wald Confidence Limits; ref.—reference group.

**Table 6 nutrients-14-02697-t006:** Declaration on choice of the food product depending on the available information indicating the sugar level.

Information on the Label	1 = Yes	2 = No	*p*-Value
Zero sugar	2.9	1.7	0.0111
No added sugars	3.3	2.2	0.0088
Sugar free	3.2	3.0	0.7143
Reduced sugar content	3.2	2.0	0.048
Contains only naturally occurring sugars	4.2	2.2	0.0005

Student’s *t*-test for independent groups.

**Table 7 nutrients-14-02697-t007:** Characteristics/perceptions of consumers aware of the consequences of high sugar consumption.

Items	1 = Yes	2 = No	*p*-Value
Young adults	2.9	2.7	0.7997
People knowledgeable about nutrition	4.5	1.7	<0.0001
Physically active people	4.3	3.0	0.0113
Busy people	2.6	2.7	0.8287
People with health problems	3.6	1.5	0.0003
Professionally active people	3.1	3.0	0.798
Elderly people	2.8	2.0	0.1565
People seeking food novelties	3.9	2.7	0.0373
Wealthy people	3.4	2.2	0.0421
People who enjoy cooking	3.6	1.5	<0.0001
Single people	3.1	2.2	0.0932
Convenience-oriented people related to the preparation of a meal	2.7	3.0	0.5605
People looking for bargains	2.5	2.5	0.9959
Families with children	3.1	3.5	0.4864
People who care in a special way about their health	4.2	1.7	<0.0001

Student’s *t*-test for independent groups.

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
