# Peer review of "Evaluation of Food Labelling the Products with Information Regarding the Level of Sugar: A Preliminary Study"

_nutrients, 2022, doi:10.3390/nu14132697_

Round 1
Reviewer 1 Report
Thanks to the editor for the invitation. In this study, the authors aim to find out the reasons for looking for selected information on a food label with a particular emphasis on selected information about sugar. This is an interesting study, however, maybe the scope of this research is not entirely in my field, I did not fully understand the purpose of this study. Please see my comments below:
1, Please show some important results with specific value and P-value.
2, Some words that are too absolute should be avoided, for example, “find out”, similar words rarely appear in scientific articles.
3, It is better to re-arrange the discussion section. The conclusion part is usually at the end pf discussion section.
Author Response
- Reviewer
Comment:
Thanks to the editor for the invitation. In this study, the authors aim to find out the reasons for looking for selected information on a food label with a particular emphasis on selected information about sugar. This is an interesting study, however, maybe the scope of this research is not entirely in my field, I did not fully understand the purpose of this study. Please see my comments below:
1, Please show some important results with specific value and P-value.
2, Some words that are too absolute should be avoided, for example, “find out”, similar words rarely appear in scientific articles.
3, It is better to re-arrange the discussion section. The conclusion part is usually at the end pf discussion section.
Reply:
Authors would like to thank the Reviewer for considering this study as an interesting. We greatly appreciate the Reviewer’s efforts in careful review of the paper and the valuable suggestions offered. We have included the following specific comments of Reviewer into the revised version of manuscript. All the corrections are included ‘in yellow’.
- We have rephrased the aim of our paper in order to make it more understandable.
- When it comes to specific value and p-value, we indicated them previously (tab. 4 and 5), however maybe the method of marking was not clear enough, therefore we have corrected it in order to make it clearer.
- When it comes to the discussion session, we have made some minor corrections (pp. 11, lines 380-388) as well as we have expanded the limitation section (pp. 12, lines 455-463).
Reviewer 2 Report
Introduction:
It remains to justify more precisely why it was decided to study the consumption of sugar only and other nutrients such as salt and sugar were not selected.
Methodology
It is missing to indicate how the questionnaire was elaborated, was some type of validation of the questionnaire carried out? How the questionnaire questions were selected. For example, in relation to the physical activity question “I am physically active”, how do the participants know the reference to being physically active? What would happen if someone who does not perform physical activity also checked labels?
What would happen in the event that someone does not feel identified with any answers? Do you have to choose any option necessarily?
In discussion:
It is said that there is a high consumption of sugar, but there would be a lack of data at the global level and at the country level to know how much sugar is consumed. How does high sugar consumption affect the population, are there differences by age groups? The cause of this high consumption are processed and ultra-processed products? Is there any data on the prevalence of consumption of processed and ultra-processed products? In addition to labeling, has the population been educated on this issue? What could the authors propose to improve the situation of sugar consumption?
In Institutional Review Board Statement: Referring to this research the ethical approval was not required. Please justify the reason
Author Response
2 Reviewer
- Comment: Introduction:
It remains to justify more precisely why it was decided to study the consumption of sugar only and other nutrients such as salt and sugar were not selected.
Reply:
We greatly appreciate the Reviewer’s efforts in careful review of the paper and the valuable suggestions offered. We have included the following specific comments of Reviewer into the revised version of manuscript. All the corrections are included ‘in yellow’.
Regarding the first comment, we have corrected this aspect in the introduction section in order to explain in details the problem concerning the high sugar consumption. Furthermore, in this study we focused only on ‘sugar’ in order to make the topic more obvious/clear for our respondents.
- Comment:
Methodology
It is missing to indicate how the questionnaire was elaborated, was some type of validation of the questionnaire carried out? How the questionnaire questions were selected. For example, in relation to the physical activity question “I am physically active”, how do the participants know the reference to being physically active? What would happen if someone who does not perform physical activity also checked labels?
What would happen in the event that someone does not feel identified with any answers? Do you have to choose any option necessarily?
Reply:
The questionnaire was elaborated as the own survey questionnaire/the original survey questionnaire. Actually, we have not carried out the validation. However, we included the description of the questionnaire in order to make the used tool more clear for the final reader (pp. 3, lines 11-119).
Regarding the question on physical activity, in general it referred to participants’ perception of the level of activity. Respondents had got the opportunity to access this level using 5-point scale (where 1 - I agree least and 5 - I agree most). We have also used this kind of question in the previous research among Polish consumers and from our point of view, it is very useful as well as understandable for the respondents.
- Comment:
In discussion:
It is said that there is a high consumption of sugar, but there would be a lack of data at the global level and at the country level to know how much sugar is consumed. How does high sugar consumption affect the population, are there differences by age groups? The cause of this high consumption are processed and ultra-processed products? Is there any data on the prevalence of consumption of processed and ultra-processed products? In addition to labeling, has the population been educated on this issue? What could the authors propose to improve the situation of sugar consumption?
Reply:
Thank you for this fruitful suggestion. Referring to the level of sugar consumption we have included the section on this aspect in the introduction section in order to underline this problem (pp. 4 lines 70-89). Moreover, we have included some information in the limitation section in order to explain in details the complexity of the current situation regarding the educational campaigns as well as the legislative process in Poland.
- Comment:
In Institutional Review Board Statement: Referring to this research the ethical approval was not required. Please justify the reason.
Reply:
During the process of conducting the research we meet the requirements indicated in ESOMAR (European Society for Opinion and Marketing Research) code with the particular emphasis regarding our respondents’ rights. Therefore, the respondents could response in their spare time (in the most convenient time from their point of view). Moreover, the study was planned directly among adults who expressed their willingness to participate in the study. For this purpose appropriate information was provided to survey participants, allowing them to decide their participation in this study. Furthermore, the respondents could quit filling in the questionnaire in any time. Additionally, we did not collect any information referring to their BMI (the weight, the height).
Round 2
Reviewer 1 Report
The response looks good.